# BcWRKY33A Enhances Resistance to *Botrytis cinerea* via Activating *BcMYB51-3* in Non-Heading Chinese Cabbage

**DOI:** 10.3390/ijms23158222

**Published:** 2022-07-26

**Authors:** Huiyu Wang, Yushan Zheng, Dong Xiao, Ying Li, Tongkun Liu, Xilin Hou

**Affiliations:** 1State Key Laboratory of Crop Genetics & Germplasm Enhancement, Key Laboratory of Biology and Genetic Improvement of Horticultural Crops (East China), Ministry of Agriculture and Rural Affairs of China, Engineering Research Center of Germplasm Enhancement and Utilization of Horticultural Crops, Ministry of Education of China, Nanjing Agricultural University, Nanjing 210095, China; 2019204023@njau.edu.cn (H.W.); 2021204027@stu.njau.edu.cn (Y.Z.); dong.xiao@njau.edu.cn (D.X.); yingli@njau.edu.cn (Y.L.); 2Nanjing Suman Plasma Engineering Research Institute, Nanjing Agricultural University, Nanjing 210095, China

**Keywords:** *Botrytis cinerea*, BcWRKY33A, BcMYB51-3, IGSs’ biosynthetic genes

## Abstract

The transcription factor WRKY33 is a vital regulator of the biological process of the necrotrophic fungus *Botrytis cinerea* (*B. cinerea*). However, its specific regulatory mechanism remains to be further investigated. In non-heading Chinese cabbage (NHCC, *Brassica campestris* (syn. *Brassica rapa*) ssp. *Chinensis*), our previous study showed that BcWRKY33A is induced not only by salt stress, but also by *B. cinerea* infection. Here, we noticed that BcWRKY33A is expressed in trichomes and confer plant defense resistance. Disease symptoms and qRT-PCR analyses revealed that *BcWRKY33A*-overexpressing and -silencing lines were less and more severely impaired, respectively, than wild type upon *B. cinerea* treatment. Meanwhile, the transcripts’ abundance of indolic glucosinolates’ (IGSs) biosynthetic genes is consistent with plants’ *B. cinerea* tolerance. Identification and expression pattern analysis of BcMYB51s showed that BcMYB51-3 has a similar trend to BcWRKY33A upon *B. cinerea* infection. Moreover, BcWRKY33A directly binds to the *BcMYB51-3* promoter, which was jointly confirmed by Y1H, dual-LUC, and EMSA assays. The importance of MYB51, the homolog of BcMYB51-3, in the BcWRKY33A-mediated *B. cinerea* resistance was also verified using the TRV-based VIGS system. Overall, our data concludes that BcWRKY33A directly activates the expression of *BcMYB51-3* and downstream IGSs’ biosynthetic genes, thereby improving the *B. cinerea* tolerance of NHCC plants.

## 1. Introduction

WRKY proteins, which share a highly conserved WRKY domain, are common DNA-binding transcription factors (TFs) that have been extensively studied in plants’ responses to environmental stimuli and pathogen attack [1]. In recent decades, numerous studies have suggested that large WRKY family members participate in defense stress responses and play vital roles in the resistance pathway. *PIWRKY65* from *Paeonia lactiflora* performed a positive role in regulating the jasmonic acid (JA) and salicylic acid (SA) signaling pathways in response to the infection of *Alternaria tenuissima* [2]. In wild tomato (*Solanum habrochaites*) LA1777 seedlings, the silencing of the *SHWRKY41* gene reduced the production of H_2_O_2_ following *Oidium neolycopersici* (*On*-lz) infection, indicating its important role in resistance to the *On*-lz strain [3]. Plants respond to necrotrophic pathogens, including *Botrytis cinerea* (*B. cinerea*), via generally activating the JA signaling pathway. The transgenic *Arabidopsis* overexpressing *PtrWRKY18* and *PtrWRKY35* showed high expression levels of *pathogenesis-related gene 3* (*PR3*) and *plant defensin gene v1.2* (*PDF1.2*), two marker genes in the JA signaling pathway, and exhibited higher tolerance to *B. cinerea* relative to wild-type (WT) plants [4]. Given that WRKY TFs perform functions in plant resistance to various pathogens, there is an urgent need to understand specific molecular regulatory mechanisms between WRKY TFs and pathogens. For instance, a previous study showed that AtWRKY33 is the key regulator in the defense against *B. cinerea* in *Arabidopsis*; its mutant *wrky33* is more susceptible after pathogen infection [5,6]. Further research revealed that sigma factor binding proteins (SIBs) were located in the nucleus and were sharply induced after infection with *B. cinerea*. As the proteins interact with WRKY33, SIBs activate the function of WRKY33 in plant defense resistance [7]. Recently, another study provided evidence that the C3H14 protein mediates the plant resistance to *B. cinerea* through WRKY33 signaling in *Arabidopsis* [8]. However, the detailed molecular mechanism between WRKY33 and necrotrophic pathogen *B. cinerea* is not fully understood.

*B. cinerea* is a broad-host and destructive fungal pathogen, which always causes a large number of gray mold layers on the surface of plants, resulting in leaf drop, as well as flower and fruit rot [9]. In the humid environment and cool temperature (15–23 °C), its conidia widely exist in the air, which can not only infect field crops, but also the most common pathogen that causes huge losses in the postharvest stage of plants [10]. Recent studies showed that plants resist *B. cinerea* through multiple strategies. For instance, after treatment with SO_2_, plants perform increased disease resistance against *B. cinerea* via regulating the activities and transcripts of defense-related enzymes and genes (such as *PAL*, *PPO*, *PR2*, *PR3*, and *CHI*) in *Arabidopsis* [11]. AtMYB46, which was described to be involved in cell wall biosynthesis in the stem, could regulate the expression of the *Ep5c* gene, and integrate defense-related signaling pathways and cell wall programming to modulate the *Arabidopsis* defense against *B. cinerea* [12]. Lately, some studies explored the relationship between *B. cinerea* and glucosinolates (GSs). Researchers discovered that *B. cinerea* has differential susceptibility to GSs [13]. Similarly, the metabolic analysis showed many secondary metabolites enriched in the GSs’ biosynthetic pathway in *B. cinerea*-infected roses [14]. Nonetheless, the role of GSs in plant resistance against *B. cinerea* is largely unexplored.

GSs are secondary metabolites that are mainly found in Brassicaceae vegetables [15,16,17]. Under the action of the endogenous myrosinase enzyme, GSs are always hydrolyzed to various degradation metabolites, which release characteristic flavors that inhibit the growth of microorganisms and act as a deterrent to certain insects and pathogens [18]. According to the different amino acid precursors and side-chain molecular structures, GSs can be divided into indolic glucosinolates (IGSs), aliphatic glucosinolates (AGSs), and aromatic glucosinolates (ARGSs) [19,20]. Among them, IGSs and AGSs are two main types of GSs in Brassicaceae and *Arabidopsis* [21]. IGSs are mainly distributed in the vegetative tissue of plants, including rosette leaves and roots, whereas AGSs are mainly found in reproductive tissue, such as seeds, flowers and siliques [22]. Previous studies suggested that the breakdown product of 4-methoxyindole-3-ylmethyl glucosinolate (4MI3G) performed effective functions in the process of plants defending themselves from fungi and pathogens [23]. Meanwhile, glucobrassicin, a compound of IGSs, conferred stronger disease resistance than AGS compounds in kale (*Brassica oleracea*) [24]. Furthermore, hydrolysates of IGSs are involved in plant resistance to *B. cinerea* [25]. Thus, tryptophan-derived IGSs and their degradation products, as well as representative disease-resistant secondary metabolites, contribute to the process of plant resistance against pathogenic bacteria and are required for plant immunity [23,26].

The biosynthetic pathway of IGSs contains multiple genes and enzymes, and this synthetic process is variable and complex [17]. Three R2R3-MYB TFs (MYB34, MYB51, and MYB122) activate promoters of IGSs’ biosynthetic genes, resulting in the accumulation of IGSs [27]. Among them, the overexpression of *MYB51* led to the specific accumulation of IGSs, whereas the overexpression of *MYB34* and *MYB122* led to the accumulation of IGSs with a high-auxin phenotype in *Arabidopsis* [27]. MYB34 and MYB51 control the expression of IGSs’ biosynthetic genes in the roots and shoots, respectively, mediating the biosynthesis of IGSs for plant disease resistance, whereas MYB122 only plays an accessory role in this process [28]. As the key regulator of IGSs’ biosynthesis, MYB51 is involved in many resistance processes, such as pathogen immunity and resistant protein activation. Under the treatment of 22 amino acid flagellin peptides (flg22), some WRKY TFs (including WRKY33) were induced in an MYB51-dependent manner [29], but the WRKY33-MYB51 signal crosstalk that regulates plant resistance against *B. cinerea* remains obscure.

In non-heading Chinese cabbage (NHCC, *Brassica campestris* (syn. *Brassica rapa*) ssp. *Chinensis*), the roles of BcWRKY33A in salt tolerance were partially investigated in our previous study [30]; however, the function of BcWRKY33A in disease resistance remains unknown. Here, we first discovered that BcWRKY33A is specifically expressed in the trichomes of leaves. Using physiological and molecular biological assays, the characteristics, functions, and relationship of BcWRKY33A and BcMYB51-3 in terms of *B. cinerea* resistance are explored. BcMYB51-3 lies downstream of BcWRKY33A and regulates the transcriptional abundance of IGSs’ biosynthetic genes, thereby improving the *B. cinerea* resistance of the NHCC plants.

## 2. Results

### 2.1. BcWRKY33A Expresses in Trichomes and Mediates Plant Resistance against B. cinerea Infection

Given that the established findings of WRKY33 have profound effects on *B. cinerea* resistance [7,8], the response of BcWRKY33A upon *B. cinerea* infection was analyzed to understand its role in plant defense. As shown in Figure 1a,b and in [30], transgenic *Arabidopsis* overexpressing *BcWRKY33A* (*35S:BcWRRKY33A*, # 1, # 4, Appendix A) showed milder symptoms relative to WT upon *B. cinerea* infection at 2 days post inoculation (dpi). Then, *BcWRKY33A*-silencing (pTY-BcWRKY33A) and pTY control plants were established by the VIGS system (Appendix A). Unlike the resistant phenotype of *BcWRKY33A*-overexpressing plants, pTY-BcWRKY33A exhibited more severe symptoms compared to the pTY control after *B. cinerea* infection at 2 dpi (Figure 1c,d). The phenotypic analysis of *BcWRKY33A*-overexpressing and silencing plants revealed that BcWRKY33A exhibits a positive role in plant resistance defense, like its homolog WRKY33 [31]. After exploring the spatial expression of BcWRKY33A, we found that GUS signaling controlled by the *BcWRKY33A* promoter (pBcWRKY33A) was mainly detected in the roots and trichomes of leaves (Figure 1e). Our previous data revealed that BcWRKY33A expressed in the roots enhanced salt tolerance in plants [30]. The important role of *Arabidopsis* trichomes in the plant-environment was well-established [32], so BcWRKY33A expressed in trichomes facilitated our investigation of its other unknown functions in plant defense. After the observation of trichomes on WT and *35S:BcWRKY33A* lines (Figure 1f), the number and density of trichomes per leaf were measured by methods used in previous studies [33,34]. Interestingly, the abundance of *BcWRKY33A* had no effect on the trichome number or density of ten-day-old *Arabidopsis* leaves (Figure 1g,h), hinting that BcWRKY33A contributed to the synthesis of some disease-resistant metabolites in the trichomes.

### 2.2. BcWRKY33A Modulates the Expression of IGSs’ Biosynthetic Genes

Trichomes are considered to be the main place for the synthesis of IGSs [32], and the role of the trichome-specific expression gene *BcWRKY33A* in the biosynthesis of IGSs was investigated. Existing studies that indicated the first step in the biosynthesis of IGSs was the conversion of tryptophan to indole-3-acetaldoxime (IAOx), catalyzed by CYP79B2 (Appendix A) [28], which was significantly increased in *35S:BcWRKY33A* compared to WT plants (Figure 2a). Next, the conversion of IAOx to indole-3-yl-methyl-GSL (I3M) was regulated by genes downstream of IAOx, including *CYP83B1*, *GSTF9*, *UGT74B1*, and *SOT16*, etc. (Appendix A) [28]. Most of these downstream genes were also upregulated in *35S:BcWRKY33A* plants (Figure 2a). Then, the conversion of I3M to 4MI3G and 1MI3G with some secondary modifications was mediated by CYP81F1/F2/F3/F4, IGMT1/2, etc. (Appendix A) [28], the transcript abundance of *IGMT1* and *IGMT2* was induced to a larger extent in *35S:BcWRKY33A* plants compared to WT (Figure 2a). In contrast, the above-mentioned IGSs’ biosynthetic genes are generally reduced in pTY-BcWRKY33A with a low expression level of *BcWRKY33A* relative to the pTY control (Figure 2b). Obviously, BcWRKY33A reprogramed the expression of many IGSs’ biosynthetic genes; however, the underlying regulatory mechanism requires further study.

### 2.3. Identification of BcMYB51s Proteins in NHCC

Previous research showed that MYB51 is a defense-related gene that may interact with WRKY33, which is involved in the plant response to pathogen attacks [27,28,29]. To understand their relationship with NHCC plants, the homologs of MYB51, three putative BcMYB51-like proteins, BraC08g029560, BraC06g013880, and BraC09g065110, were found in the NHCC genome, and they were named BcMYB51-1, -2, and -3 [35]. As shown in Figure 3a, the three BcMYB51-like proteins share similar gene structures with AtMYB51, with three exons and two introns in the full-length cDNA. A comparison of the amino acid sequences showed that all BcMYB51s have high identities and contain two conserved SANT motifs, which are also named MYB-like DNA binding domains (Figure 3b). The genome triplication event occurred in most Brassicaceae crops during the evolutionary history of *Arabidopsis* [35]. The phylogenetic tree, including MYB51-like proteins in six Brassicaceae species, showed that BcMYB51-3 shared a closer relationship with BcMYB51-1 compared to BcMYB51-2, although they were all derived and differentiated from AtMYB51 (Figure 3c). After obtaining the recombinant plasmids 35S:BcMYB51-1(2/3)-GFP, the subcellular localization of BcMYB51s proteins showed that the GFP signals were overlapped with histone H2B-RFP, which expressed as a nuclear marker [36], indicating the three BcMYB51s proteins located in the nucleus (Figure 3d).

### 2.4. The Expression Patterns Analysis of BcMYB51s

To understand the expression patterns of BcMYB51s, the NHCC ‘suzhouqing’ seedlings were treated with salt (100 mM, NaCl), cold (4 °C), heat (37 °C) treatment and *B. cinerea* infection, the temporal expressions of *BcMYB51s* were analyzed via qRT-PCR. Within 12 h of salt treatment, the expression levels of *BcMYB51-1* and *BcMYB51-3* hardly changed and only decreased at 12 h in the leaves. Under the same conditions, the expression of *BcMYB51-2* reached a peak at 6 h, then decreased between 6 and 12 h in the leaves (Figure 4a). After the low-temperature treatment, the abundance of *BcMYB51-1* and *BcMYB51-3* began to continuously decline in leaves, whereas the abundance of *BcMYB51-2* fluctuated over 12 h. Meanwhile, in the root tissue, *BcMYB51-1* and *BcMYB51-3* decreased slightly after acute induction, but maintained high expression levels relative to 0 h, while *BcMYB51-2* showed a completely different response (Figure 4b). During the high-temperature treatment, *BcMYB51s* significantly increased in leaves after 3 h, but rapidly responded and remarkably decreased in roots (Figure 4c). Focusing on the disease-resistance properties of MYB51, we also checked the expression of *BcMYB51s* when infected with *B. cinerea*. As expected, the expressions of *BcMYB51-1* and *BcMYB51-3* were upregulated after infection, while *BcMYB51-2* appeared to be unaffected (Figure 4d).

To further investigate the regulatory mechanism of BcMYB51s, the three 2000 bp promoters of *BcMYB51s* (pBcMYB51-1, pBcMYB51-2, and pBcMYB51-3) were analyzed using PLANTCARE databases [37]. As we know, WRKY TFs usually regulate the expression of target genes via W-box *cis*-elements [38,39,40]. Notably, both pBcMYB51-1 and pBcMYB51-3 have two W-boxes, whereas pBcMYB51-2 only has one, suggesting that in some mutations, indels may have occurred during the evolutionary process (Appendix A). Additionally, promoter analysis showed that there were many light-responsive and phytohormone-related responsive elements in the promoters of *BcMYB51s*, and functional prediction indicted their multifunctional capabilities (Appendix A).

### 2.5. BcWRKY33A Binds to the W-Boxes in the BcMYB51-3 Promoter

Tissue-specific expression analysis showed that MYB51 is present in *Arabidopsis* trichomes and involved in IGSs’ biosynthesis [32]. Interestingly, pBcWRKY33A:GUS expressed in trichomes and BcWRKY33A mediated the expression of IGSs’ biosynthetic genes (Figure 1c and Figure 2). After the treatment of *B. cinerea*, *BcMYB51-3* showed a similar trend to *BcWRKY33A* in NHCC (Figure 5a), as did their homologs in *Arabidopsis* (Figure 5b), hinting that BcMYB51s may interact with BcWRKY33A to jointly respond to *B. cinerea* with an unknown mechanism.

Based on these data, it is reasonable to speculate that BcWRKY33A can activate *BcMYB51*s by binding to their promoters. To understand the relationship between BcWRKY33A and BcMYB51s, the Y1H assay was used to check their interaction. Results showed that only nBcWRKY33A but not cBcWRKY33A could bind to the promoter fragment of *BcMYB51-1* (pBcMYB51-1, -2000 to -322, without the 5′UTR region). In contrast, neither nBcWRKY33A nor cBcWRKY33A could bind to the promoter fragment of *BcMYB51-2* (pBcMYB51-2, -2000 to -432, without the 5′UTR region). Regarding the promoter fragment of *BcMYB51-3* (pBcMYB51-3, -2000 to -410, without the 5′UTR region), nBcWRKY33A interacted weakly with pBcMYB51-3, while cBcWRKY33A interacted strongly with pBcMYB51-3 (Figure 6a). Combined with the above data that BcMYB51-3 was the most obviously induced by *B. cinerea* (Figure 4d), it was selected for further study. The analysis of the dual-luciferase (LUC) reporter suggested that BcWRKY33A could activate the expression of *BcMYB51-3* (Figure 6b,c). To confirm whether the interaction between BcWRKY33A and pBcMYB51-3 is mediated by the two predicted W-boxes found in the results above (Appendix A), two predicted W-boxes and their surrounding sequences were designed as Probes 1 and 2 for the EMSA assay (Appendix A). The results show that BcWRKY33A can directly bind to Probes 1 and 2, and their bind shifts competed with unlabeled probes (Competitor 1 and 2), suggesting that the physical binding between the BcWRKY33A protein and two predicted W-boxes in pBcMYB51-3 is specific (Figure 6d).

### 2.6. The Homolog of BcMYB51-3 Is Required for BcWRKY33A-Mediated Resistance to B. cinerea in Transgenic Arabidopsis

Next, the important role of MYB51, the homolog of BcMYB51-3, in the BcWRKY33A-mediated resistance against *B. cinerea* was checked. The TRV/*35S:BcWRKY33A* and TRV-MYB51/*35S:BcWRKY33A* lines were obtained using the TRV-based VIGS system in the transgenic *Arabidopsis*
*35S:BcWRKY33A* (# 4) line (Figure 1) [41]. To verify the VIGS system is effective in # 4, the silencing of the *Arabidopsis Phytoene desaturase* (*PDS*) gene was observed as the positive control. In our data, TRV-PDS/*35S:BcWRKY33A* lines produced a typical white color in plants (Appendix A). Meanwhile, TRV-based VIGS did not exert an effect on the expression *BcWRKY33A* in various silenced lines (Appendix A). In the TRV-MYB51/*35S:BcWRKY33A* lines, the silencing of MYB51 was verified using qRT-PCR (Figure 7a), as expected, the transcript abundance of some IGSs’ biosynthetic genes, including *CYP79B2*, *CYP83B1*, *UGT74B1*, and *SOT16*, were also reduced in TRV-MYB51/*35S:BcWRKY33A* compared to TRV/*35S:BcWRKY33A* lines. However, the expression of *IGMT1* and *IGMT2* hardly underwent alterations in TRV/*35S:BcWRKY33A* or TRV-MYB51/*35S:BcWRKY33A* lines (Figure 7b). The further observation and calculation of the lesion size of leaves infected with *B. cinerea* indicated that the silencing of *MYB51* reduced the resistance of *35S:BcWRKY33A* transgenic *Arabidopsis* against *B. cinerea* (Figure 7c,d).

## 3. Discussion

WRKY33 TF, a member of the WRKY family, has been extensively studied as a multifunctional regulator of plants involved in various abiotic and biotic stresses. Recently, researchers found that WRKY33, together with WRKY12 and RAP2.2, formed a feedback regulation system in response to the hypoxia stress due to submergence [38]. Under the conditions of reactive oxygen species (ROS) accumulation, the WRKY33-PIF4 loop is one of the sophisticated strategies of plants to cope with stress [42]. Similarly, our previous study showed that BcWRKY33A interacts with BcHSFA4A to increase salt-related gene expression, thereby enhancing salt tolerance in NHCC plants [30]. Obviously, understanding the molecular mechanism between WRKY33 and its interacting proteins or downstream genes is the key to exploring its versatility. In our data, plants overexpressing and silencing BcWRKY33A exhibited diametrically opposite resistance to *B. cinerea*, indicating the positive defense ability of WRKY33 and BcWRKY33A (Figure 1a–d) [30,31]. Meanwhile, we first noticed that pBcWRKY33A was expressed in trichomes by GUS staining and was consistent with the spatial expression of MYB51 (Figure 1e) [32], which was not found in previous studies. Interestingly, BcWRKY33A is not involved in affecting the number and density of trichomes (Figure 1f–h), but rather regulates the expression of IGSs’ biosynthetic genes, which is highly consistent with the level of disease resistance (Figure 1a–d and Figure 2), suggesting that some IGSs’ biosynthetic genes participated in a BcWRKY33A-mediated *B. cinerea* resistance process.

MYB proteins are widely distributed and functionally diverse in all eukaryotes [43]. These members of the MYB family can be categorized based on protein structure (1R-, R2R3-, 3R-, and 4R-MYB proteins) [44]. Among them, R2R3-MYB types are superior in numbers in plants, including our BcMYB51s. The importance of MYB TFs in the various programs of plants was shown in previous data, including but not limited to cell morphogenesis and differentiation [45,46], trichome development [47], hormone responses [48], abiotic and biotic stress [49,50], and light response [51]. Regarding MYB51 and its homologs, *AtMYB51* was up-regulated after wound treatment [52]. In the functional prediction of promoters of *BcMYB51s* (Appendix A, Appendix A), we found many generic elements including AE-box, AT1-motif, chs-CMA1/2a, GA-motif, GATA-motif, G-Box, and TCT-motif, indicating that *BcMYB51s* may respond to light signals [53,54,55]. Some phytohormone-related responsive elements were also found, such as AuxRR-core, CGTCA-motif, P-box, TATC-box, and TGAVG-motif, suggesting that plant hormones are involved in the regulation of *BcMYB51s* [56,57]. Understandably, pBcMYB51-3 has a unique LTR element (involved in low-temperature responsiveness), which is consistent with the highest expression of *BcMYB51-3* induced by low temperature (Figure 4b). At the same time, three *BcMYB51s* were also induced by salt, cold, and heat treatment with various trends (Figure 4); the underlying functions of BcMYB51s upon exposure to abiotic stresses also warrants further study.

Additionally, among three BcMYB51s, phylogenetic analysis showed that BcMYB51-3 shared a close relationship with BcMYB51-1 compared with BcMYB51-2 (Figure 3c); the expression pattern analysis after stress treatment also suggested that BcMYB51-1 and BcMYB51-3 displayed similar induction trends compared to BcMYB51-2 (Figure 4). These findings may be related to the whole-genome triplication event that occurred in the evolution of Cruciferae crops [58]. Similarly, three BREVIS RADIX (BRX) members, BrBRX.1, BrBRX.2, and BrBRX.3 evolved from Arabidopsis BRX (AtBRX) to control the leaf-heading phenotype of Chinese cabbage (*Brassica rapa*), with totally different expressions [58]. Although our study mainly focused on the most significant differentially expressed *BcMYB51-3* upon *B. cinerea* infection, whether BcMYB51-1 has redundant or unique functions with BcMYB51-3 upon *B. cinerea* infection requires further verification.

The necrotrophic pathogen *B. cinerea* spreads widely in plants, causing severe crop losses. Recent studies have shown that CCCH-type protein C3H14 enhances plant tolerance to *B. cinerea* depending on WRKY33 signaling [8]. Although WRKY33 is thought to be required for plant defense against necrotrophic fungal pathogens, including *B. cinerea*, the detailed molecular mechanism is not clear [59]. Previous studies have speculated that MYB51 interacts with other TF-dependent ethylene pathways (including WRKY33) during pathogenic immunity [29], whereas the detailed mechanism remains unclear. Our study provides evidence that BcWRKY33A mediated plant resistance to *B. cinerea* via directly binding and activating the expression of *BcMYB51-3*, and this result was jointly confirmed by Y1H, LUC, and EMSA assay (Figure 6). Under *B. cinerea* infection, *BcWRKY33A* showed an earlier and sharper increase than *BcMYB51-3* at 12 h (Figure 5a), so it is reasonable and understandable that BcMYB51-3 functions downstream of BcWRKY33A in NHCC plants. Coincidentally, a recent study revealed that WRKY33 could activate the expression of IGS-related genes, then enhance the *Arabidopsis* and Brassica crop resistance against another necrotrophic fungus, *Alternaria brassicicola* [50]. Thus, the importance of MYB51, the homolog of BcMYB51-3, in BcWRKY33A-mediated resistance to *B. cinerea* was checked (Figure 7). After silencing *MYB51* expression in *35S:BcWRKY33A* lines, the resistance of *BcWRKY33A*-overexpressing plants (TRV-MYB51/*35S:BcWRKY33A*) was impaired, with the reduced transcriptional abundance of some IGSs’ biosynthetic genes, indicating that MYB51 is required in this process (Figure 7). Taken together, our study reveals that BcWRKY33 directly binds to the promoter of *BcMYB51-3* to activate its expression, then regulates IGSs’ biosynthesis gene abundance to improve plant resistance to *B. cinerea* (Figure 8).

## 4. Materials and Methods

### 4.1. Plant Materials and Growth Conditions

*Arabidopsis thaliana* ‘Columbia-0’ ecotype and NHCC ‘suzhouqing’ were used as a control in our experiments unless stated otherwise. All plants were grown under long-day conditions (16/8 h, light/dark, respectively) in an artificial climate chamber at 22/18 °C, respectively. The *BcWRKY33A*-overexpressing *Arabidopsis* lines harbor recombinant construct 35S:BcWRKY33A-GFP, and the *BcWRKY33A*-silencing NHCC lines harbor recombinant construct pTY-BcWRKY33A. Methods of plant transformation were performed as described in our previous study [30].

### 4.2. Bioinformatics Analysis

The sequence information of BcMYB51s was obtained from the Non-heading Chinese Cabbage Database (http://tbir.njau.edu.cn/NhCCDbHubs/index.jsp, accessed on 20 May 2021) [35], and MYB51 homologs in six cruciferous plants were obtained from the Brassicaceae Database (http://brassicadb.cn/#/, accessed on 20 May 2021) [60]. Gene structure was displayed by GSDS 2.0 [61], the multiple sequence alignment of BcMYB51s and AtMYB51 was analyzed using jalview software [62], and the phylogenetic tree was derived from MEGA X software (neighbor-joining algorithm, bootstrap replications = 1000).

### 4.3. Subcellular Localization

The ORFs of *BcMYB51-1*, *BcMYB51-2*, and *BcMYB51-3* genes without termination codon were cloned and inserted into the *Nde*I-*Kpn*I-cut pRI101 vector to obtain recombinant plasmids (see the primers in Appendix A). Three constructs harboring BcMYB51s were co-injected with nuclear marker histone H_2_B-RFP into one-month-old tobacco plants [36]. The fluorescent signals were detected using laser scanning microscopes (LSM780, ZEISS, Germany) [30].

### 4.4. Infection Assay

The infection assay with *B. cinerea* was performed as described in a previous study with simple modifications [63]. In brief, *B. cinerea* used in our study was cultured and expanded on a V8 agar medium for 10 days at 22 °C under dark conditions. The *B. cinerea* were collected and suspended in a 25 mL SMB medium, followed by stirring for more than 0.5 h to release the spores. For the pathogenicity test on NHCC with detached leaves, the petiole was wrapped in damp cotton and the detached leaves were placed on wet filter paper in dishes. After inoculating each leaflet with 10-μL spore suspension droplets, the dishes were covered with lids and kept in a dark place for 48 h. For the pathogenicity test on leaves of *Arabidopsis* seedlings, each leaflet was inoculated with a 5-μL droplet and the phenotype was recorded at 2 dpi in a dark and humid environment. The V8 agar and SMB medium are shown in Appendix A.

### 4.5. Virus-Induced Gene Silencing (VIGS)

The NHCC silence lines were generated using the TYMV-based VIGS system [64]. pTY and pTY-BcWRKY33A were obtained in our previous study [30]. The TRV, TRV-PDS, and TRV-MYB51 lines in the *35S:BcWRKY33A* (# 4) background were produced using a TRV-based VIGS system with slight modifications [41]. In brief, the *PDS* and *MYB51* cDNA fragments was amplified using primers: PDS-F (5′-TTCTGCGGCGAATTTGCCTTATCAAAACG-3′), PDS-R (5′-AGAAACTCTTAACCGTGCCATCGTCATTGAG-3′), MYB51-F (5′-AGCTCGTGGACTACCAGGAA-3′), and MYB51-R (5′-TTGACGTTCATAGACCGGCG-3′). The resulting PCR products were inserted into the *Eco*RI-*Xba*I-cut TRV2 to obtain recombinant constructs (TRV-PDS and TRV-MYB51). After transformation using the *Agrobacterium tumefaciens* strain *GV3101*, pTRV1 and pTRV2 constructs were co-injected into fifteen-day-old *35S:BcWRKY33A* (# 4) seedlings with needleless 1 mL syringes. After infiltrating the entire leaves of the seedlings, they were kept in dark conditions overnight.

### 4.6. Yeast One-Hybrid Assay (Y1H)

In yeast one-hybrid assay, the insertion of the full-length BcWRKY33A protein into the pGADT7 vector may cause yeast cell death for unknown reasons. To avoid its toxic effects on yeast, we truncated the BcWRKY33A according to its amino acid sequence and motifs to nBcWRKY33A (1-158 amino acids, aa) and cBcWRKY33A (159-476 aa), the starting codon ATG was added to the 5′ end of cBcWRKY33A to ensure its normal translation (Appendix A), and they were integrated into the pGADT7 vector, referring to previous research [30]. Three promoters of *BcMYB51s* were amplified from the gDNA of NHCC ‘suzhouqing’ and the products were inserted into the pAbAi vector. After confirming the appropriate inhibition concentration of aureobasidin A (AbA), the interaction of pAbAi-pBcMYB51s and cBcWRKY33A or nBcWRKY33A was determined using the lithium acetate method (Clontech, Cat.630439) in Y1H Gold strains [30].

### 4.7. Dual-Luciferase Reporter Assay (LUC)

The 2000-bp promoter of *BcMYB51-3* was inserted into the *Pst*I-*BamH*I-cut pGreenII-0800-LUC vector as the reporter vector using a cloning kit (C112-01, Vazyme Biotech Co., Ltd., Nanjing, China). *35S:BcWRKY33A-GFP* and *35S:GFP* (control) were used as effector vectors, and relative primers are shown in Appendix A. After transformation into *GV3101* (pSoup), the mixture of effector and reporter (1:1) was injected into leaves of tobacco. D-luciferin was sprayed onto the leaves and a plant imaging system (Berthold, Night Shade LB 985) was used to record the LUC imaging. The LUC/REN was detected and measured using by a dual-luciferase reporter assay kit (DL101-01, Vazyme Biotech Co., Ltd., China) [30].

### 4.8. Electrophoretic Mobility Shift Assay (EMSA)

EMSA was applied using a chemiluminescent EMSA kit (GS009, Beyotime, China) according to the instruction. The recombinant proteins of BcWRKY33A were purified as referred to in our previous study [30]. The DNA-protein complexes were separated by 6% native polypropylene gels with TGE buffer (R23174, yuanye, China). After transfer of the bound and unbound signal bands onto nylon membranes, UV crosslinking and chemiluminescence were performed to obtain the final image.

### 4.9. Quantitative Real-Time PCR (qRT-PCR)

Total RNA was mainly extracted from leaves of plants using a TRIzaol reagent, and cDNA was synthesized using an Evo M-MLV Mix Kit with gDNA Clean for qPCR (Accurate Biotechnology (Hunan) Co., Ltd., Changsha, China). The gene relative expressions were detected using Hieff^®^ qPCR SYBR Green Master Mix (Low Rox Plus) (Cat No.11202ES08; Yeasen, Shanghai, China) on the QuantStudio^®^5 system and analyzed using the 2^−ΔΔCT^ method, normalized to *ELF4A* (AT1G80000) and *BcGAPDH* (BraC09g068080.1) for *Arabidopsis* and NHCC. All primers used in qRT-PCR are shown in Appendix A. Three technical repeats per sample and at least three biological repeats were set in our qRT-PCR experiments.

## Figures and Tables

**Figure 1 ijms-23-08222-f001:**
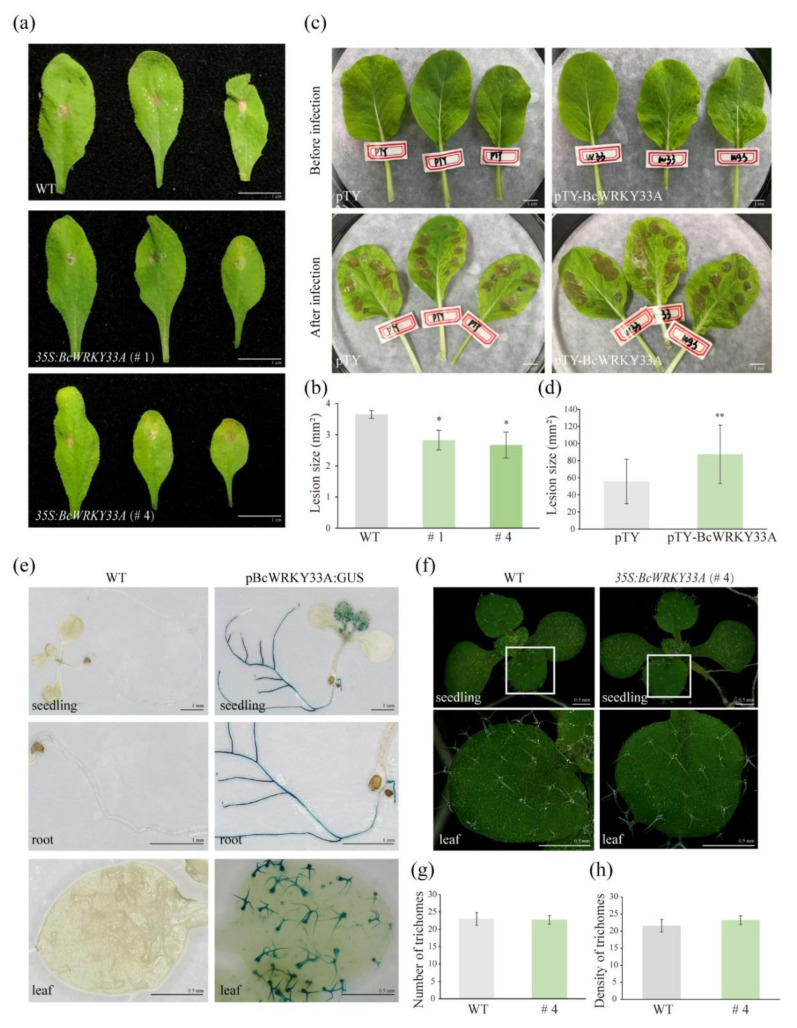
BcWRKY33A expressed in trichomes and mediated *B. cinerea* resistance. (**a**,**b**) The disease symptoms (**a**) and lesion size (**b**) of wild type (WT) and *35S:BcWRKY33A* lines (# 1, # 4) at 2 days post inoculation dpi (dpi) with a 5-μL droplet of *Botrytis cinerea* (*B. cinerea*) [30]. The average lesion size was measured based on 20 leaves from at least 6 independent plants of each line. (**c**,**d**) The disease symptoms and lesion size of pTY and pTY-BcWRKY33A lines at 2 dpi with many 10-μL droplets of *B. cinerea*. The average lesion size was measured based on each infected area on 6 leaves from 3 independent plants of each line (note: there were at least 8 droplets of *B. cinerea* on each leaf). (**e**) GUS staining of twelve-day-old WT and pBcWRKY33A:GUS seedlings. (**f**–**h**) Trichome observation (**f**), average number of trichomes (**g**), and density of trichomes (**h**) of ten-day-old WT and *35S:BcWRKY33A* (# 4) lines. After observing trichomes on rosette leaves of ten-day-old seedlings, the average number of trichomes and the density (number of trichomes per mm^2^, number/mm^2^) of trichomes per leaf were determined, and a total of 8 independent leaves (*n* = 8) were counted. Error bars represent standard deviation (SD). * *p* < 0.05, ** *p* < 0.01 (Student’s *t*-test).

**Figure 2 ijms-23-08222-f002:**
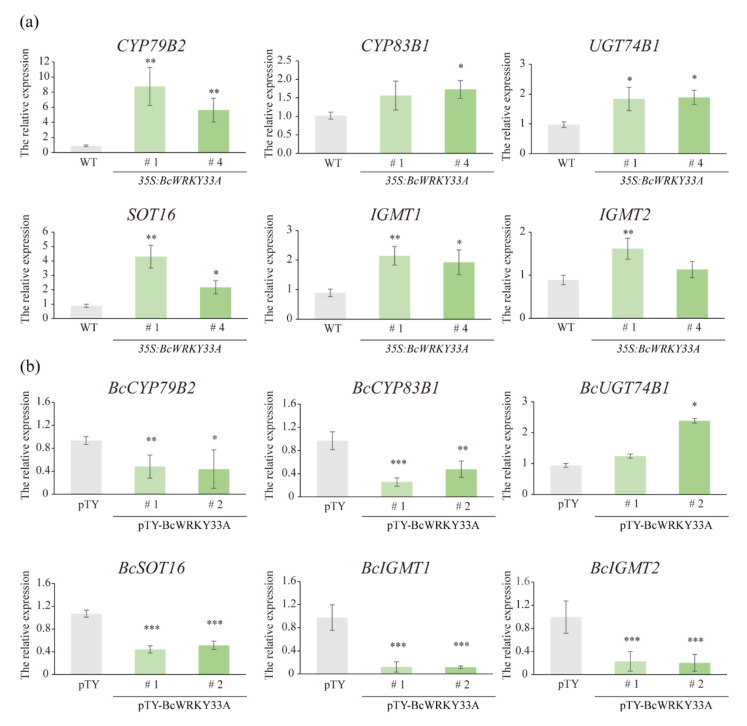
BcWRKY33A mediates the abundance of IGSs’ biosynthetic genes. (**a**) The relative expression level of IGSs’ biosynthetic genes in WT and *35S:BcWRKY33A* lines. (**b**) The relative expression level of IGSs’ biosynthetic genes in pTY and pTY-BcWRKY33A plants. Error bars represent SD. Data are means ± SD of three biological replicates. The significant differences in genes’ relative expression were measured based on Student’s *t*-test (* *p* < 0.05, ** *p* < 0.01, *** *p* < 0.001).

**Figure 3 ijms-23-08222-f003:**
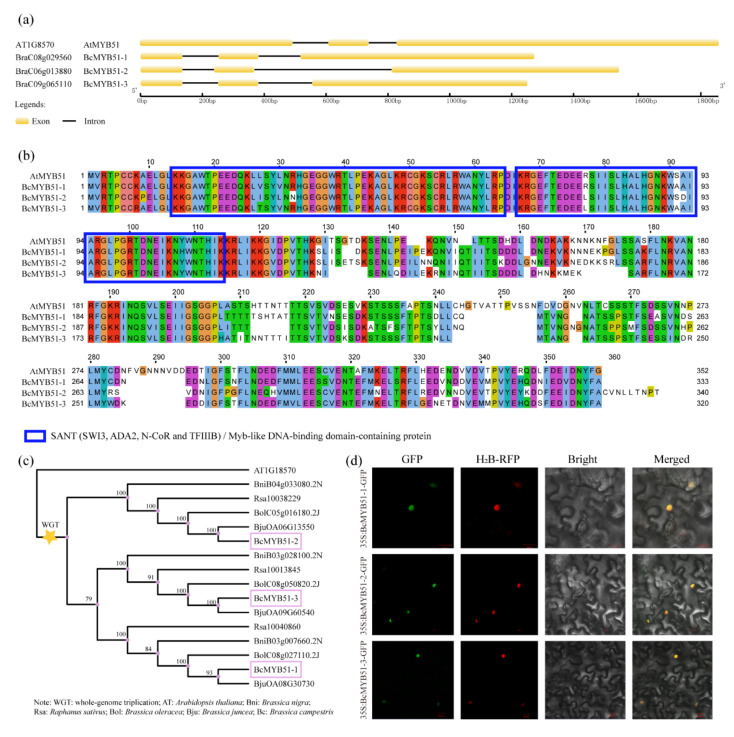
Identification of three BcMYB51-like proteins. (**a**) Gene structures of BcMYB51s and AtMYB51. (**b**) Alignment of sequences of BcMYB51s and AtMYB51. (**c**) Phylogenetic analysis of MYB51 in six cruciferous plants. WGT, whole genome triplication. (**d**) Subcellular analysis of BcMYB51s proteins. The experiments were performed twice with similar results.

**Figure 4 ijms-23-08222-f004:**
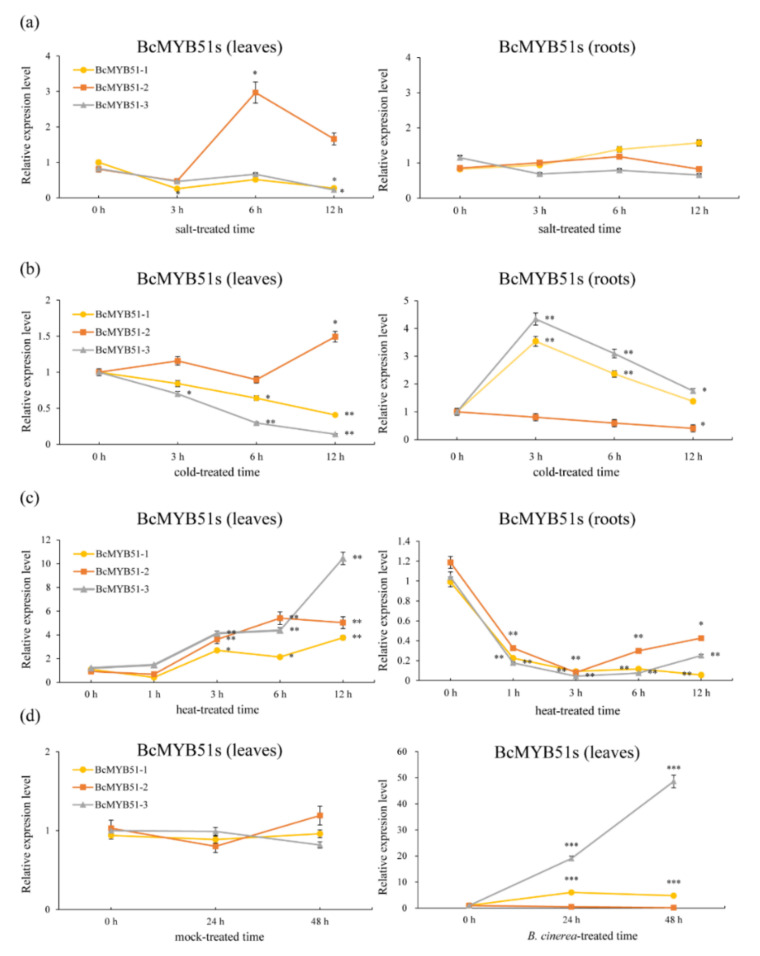
The expression patterns of *BcMYB51s* under abiotic and biotic stress. (**a**–**c**) The relative expression level of *BcMYB51s* upon salt (**a**), cold (**b**), and heat (**c**) treatment in leaves and roots of NHCC ‘suzhouqing’. (**d**) The relative expression level of *BcMYB51s* upon *B. cinerea* infection in leaves of NHCC ‘suzhouqing’. The data are the mean ± SD of three biological replicates. * *p* < 0.05, ** *p* < 0.01, *** *p* < 0.001 (Student’s *t*-test).

**Figure 5 ijms-23-08222-f005:**
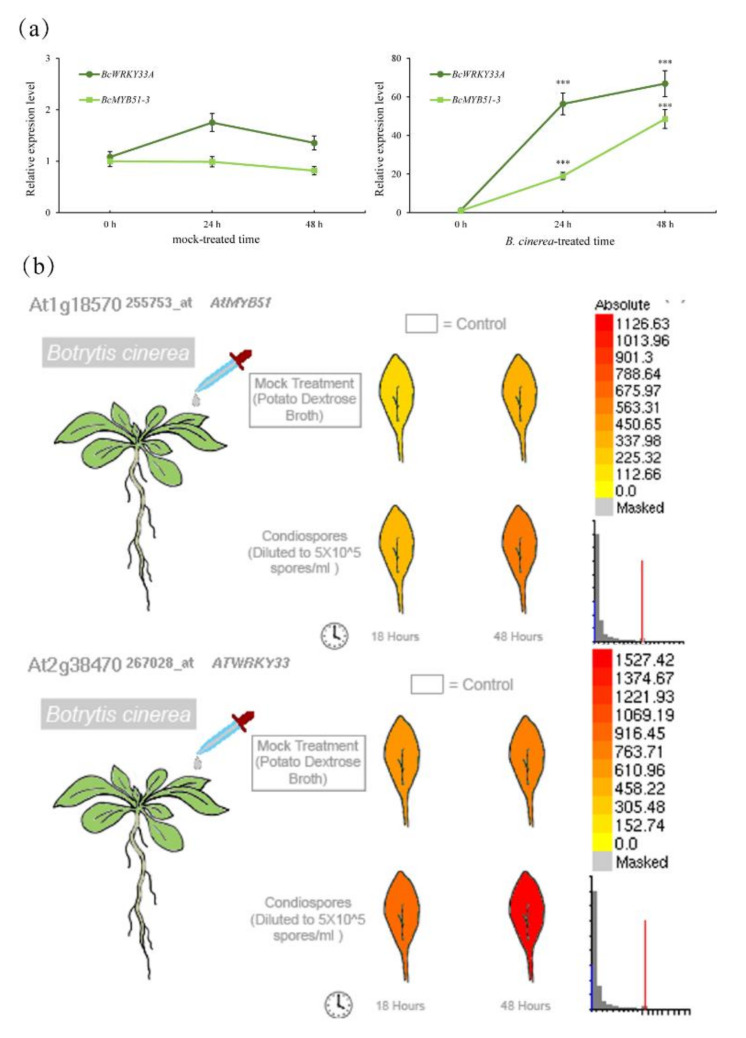
The similar expression patterns of *BcWRKY33A* and *BcMYB51-3*. (**a**) The similar expression patterns of *BcWRKY33A* and *BcMYB51-3* upon *B. cinerea* infection at 2 dpi. Error bars represent SD. The data are the mean ± SD of three biological replicates. *** *p* < 0.001 (Student’s *t*-test). (**b**)Transcriptome data showing that *B. cinerea* infection can induce the expression level of *MYB51* and *WRKY33* (data from the Arabidopsis eFP Browser, http://bar.utoronto.ca; accessed on 20 May 2021).

**Figure 6 ijms-23-08222-f006:**
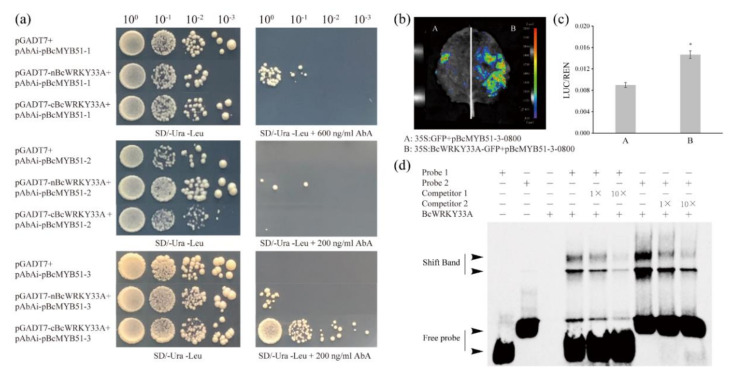
BcWRKY33A directly binds to two W-boxes in the promoter of *BcMYB51-3*. (**a**) Y1H showed that BcWRKY33A protein binds to the promoters of *BcMYB51-1* and *BcMYB51-3*. (**b**) Imaging of LUC activity showed that BcWRKY33A activates the expression of *BcMYB51-3*. (**c**) The ratio of LUC/REN of (**b**). (**d**) EMSA showed that BcWRKY33A protein binds to the two W-box elements in the promoter of *BcMYB51-3*. The experiments were performed twice with similar results. Error bars represent SD. The data are the mean ± SD of three biological replicates. * *p* < 0.05 (Student’s *t*-test).

**Figure 7 ijms-23-08222-f007:**
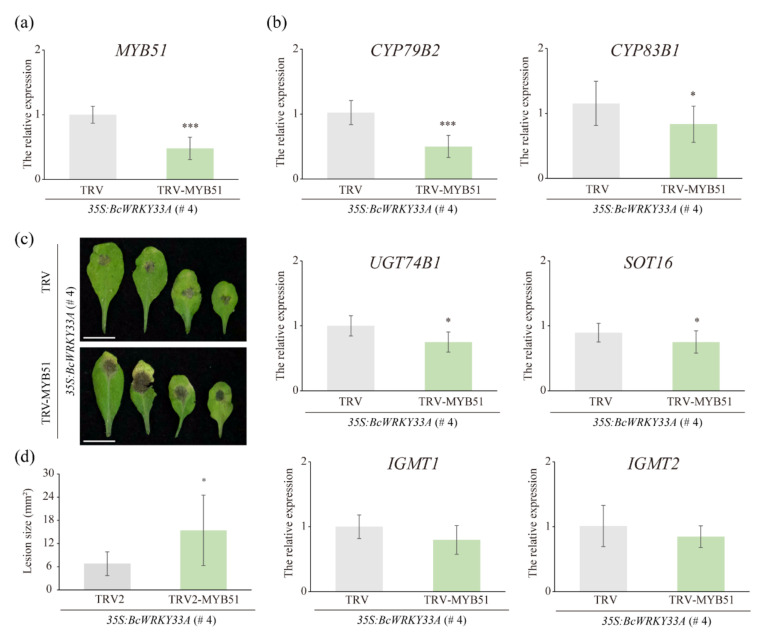
The silencing of *MYB51* reduced the *B. cinerea* tolerance of *35S:BcWRKY33A* transgenic *Arabidopsis*. (**a**,**b**) The relative expression level of *MYB51* (**a**) and IGSs’ biosynthetic genes (**b**) in TRV/*35S:BcWRKY33A* and TRV-MYB51/*35S:BcWRKY33A* lines. Data are means ± SD of four biological replicates. (**c**,**d**) The disease symptoms (**c**) and lesion size (**d**) of three-week-old TRV/*35S:BcWRKY33A* and TRV-MYB51/*35S:BcWRKY33A* lines in the background of overexpression of *BcWRKY33A* (*35S:BcWRKY33A*, # 4). Bars = 1 cm. After 2 dpi with a 5-μL droplet of *B. cinerea*, the results of a representative experiment are shown in (**c**). The average lesion size was measured based on at least 16 leaves from at least 7 independent plants of each line. Error bars represent SD. * *p* < 0.05, *** *p* < 0.001 (Student’s *t*-test).

**Figure 8 ijms-23-08222-f008:**
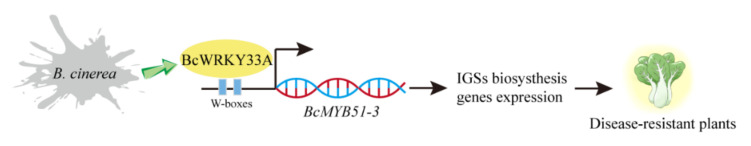
The working model of BcWRKY33A and BcMYB51-3 in the modulation of IGS’ biosynthesis gene expression to affect the interactions of plants and *B. cinerea*. The *B. cinerea* infection induced massive accumulation of BcWRKY33A, which directly binds to the W-boxes in the promoter of *BcMYB51-3*, and the activation of the transcription of *BcMYB51-3* contributed to increasing the expression of IGSs’ biosynthetic genes, thereby improving the plants’ resistance against *B. cinerea*.

## Data Availability

Not applicable.

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
