# Peer review of "BcWRKY33A Enhances Resistance to Botrytis cinerea via Activating BcMYB51-3 in Non-Heading Chinese Cabbage"

_ijms, 2022, doi:10.3390/ijms23158222_

Round 1

Reviewer 1 Report

In my opinion the manuscript entitled 'BcWRKY33A enhances resistance to Botrytis cinerea via activating BcMYB51-3 in non-heading Chinese cabbage' clearly describes the results of well-planned and correctly executed experiments.

Two minor comments are:

 - all abbreviations, such as TFs (line 33), must be explained the first time they are used;

- in my opinion it would be valuable to explain the meaning and role of PtrWRKY18 and PtrWRKY35 as well as PR3 and PDF1.2 (genes or their products) (lines: 42-43)

Reviewer 2 Report

This work describes experiment proving the interaction between the WRKY 33 TF and the promoter of MYB51-3 in Non-HeadingChinese Cabbage, thus elucidating a role of WRKY33 in plant defence against B. cinerea.

Based on clues that B. cinerea is susceptible to GSs, that IGs synthesis is regulated by MYB51, and finally that WRKY33a and MYB share a similar expression pattern, the authors postulate a connection between WRKY33a and MYB and design experiments to prove that WRKY33 regulates MYB51 upon B. cinerea infection.

The research is overall well conducted. Conclusions are sometimes, but not always, drawn by solid experiments.

Major issues:

1)

With some exception (e.g. figure 4), information about replicates -esp. biological replicates- is generally missing.

2)

Statement that "abundance of WRKY33A doesn't affect abundance and distribution of trichomes seems to be based only on qualitative data (one picture!?) 

3) It is at times unclear what was done in this work and what is a repetition of previous work, due to lack of proper citation where expected. This MUST be clear, and unequivocal. Proper references to previous work MUST be provided where needed.  It is stated (line 17) that HERE (in this work) it is shown that BcWRKY33A conferred defense resistance. This is however shown also in Figure 1b in the previous Wang's paper. 

Moreover, lines 128-130 and figure 1a propose again the experiment of Wang et al 2020, without proper citation at line 128 (As shown in figure 1a and in Wang et al 2020), and legend of figure1 .

4) The results (symptoms assessment) of pathogen infection assays are not giving clear results: the differences between lines is generally small (which is common and expected). Therefore, eventual differences need to be evaluated with a solid number of repetitions (biological replicates), especially when pictures of infected plants cannot help in appreciating such differences. 

--------

The English language must be revised and improved. There are several grammar-related errors all over the manuscript, sometimes preventing the reader to unequivocally understand the meaning of what is stated. Just to make an example:

Line 17: WRKY expressed ...and confers

Line 18: Disease symptom and qRT-PCR analyses 

Line 21: Is consistent with

Line 70: Initially explore (?)

Line 77: do you mean that GSs are specific metabolites of Brassicaceae (only produced by these species)? or rather than some of them are specific of Brassicaceae? Not clear, please rephrase.

Line 84: dominated?

Line 100: interchangeable is usually referring to at least two items. In this sentence it is referred to one item (the synthetic process). 

Line 109: which "findings of" MYB51? The link between MYB51 and WRKY (reference to Zhou, et al. 2019) is not explained clearly. Please rephrase.

Line 130: Figure S1a describes relative expression of the transgene, not symptoms. For clarity, please rephrase sentence of lines 128-130. Suggestion: place the reference of Figure S1a right after the first mentioning of the transgenic line: (35S:BcWRRKY33A, # 1, # 4, Figure S1a) 

The same stands for line 134-Figure S1b

Line 135: conserved across species?

Line 126: Perhaps substitute findings OF with findings THAT

Line 129: use "milder than wt" in place of "more mild"

Figure 1a: The fact that the lesion size in wt are more severe than those in transgenic lines is not appreciable by this image. Anyway, this experiment has been already performed in the previous published research by the same authors (Wang 2020), therefore presenting these data again here as they were novel and without proper citation is misleading.

Figure 1b: on how many plants was the assay done? How many leaves per plant? Is the n of replicates (n>3) referring to independent plants or leaves of the same plant? 

Line 143: was the number of trichomes in wt and transgenic lines quantified? 

Line 174: the T test doesn't measure relative expression. Please rephrase.

Line 182: three genes were found IN the NHCC genome, and they were named ...

Line 216: given the nature of the qRT-PCR technique, the gene expression fold change is calculated relative to a control sample (i.e. the tissue non challenged by a pathogen, in this case). If in such condition a gene is either not expressed, (expression equal to zero), or its expression is close to the detection limit of the technique, the fold change sky rockets to infinite. possibly this is the reason why such variability is detected (7-60). In this case I would suggest to replace the 7-60 fold change statement with just "upregulation".

Line 236: "previous data" is misleading. Aren't they data relative to this current work?

Line 239: please correct "strongest responded"

Line 236-241: In my opinion, the similar expression profile of BcWRKY33A and BcMYB51 represent a significant clue indicating a possible interaction. Nevertheless, a similar expression pattern is not sufficient to "suggest" interaction. Therefore, on one side I would tone down statement at line 240 (indeed, "speculate" of line 243 is appropriate). On the other hand, data of Figure S4 could be shown in a main figure of the paper (or incorporated into an existing main figure).

Line 245-247 should be moved to the methods section.

Material and methods:

Line 416: The infection assay WITH B. cinerea.

qRT-PCR: Please state how many biological replicates and technical replicates were used in all qRT-PCR experiments. Moreover, in figures reporting qRT-PCR experiments results, please state what the error bars refer to, and relative to how many replicates (missing in figure S1).

Line 425: the sentence is missing a proper verb.

Line 270: Asterisk indicates significance according to the T test.

The T test doesn't provide mean or SD.

line 282: similar phenotype in what sense? The picture is not of a sufficient quality to discriminate eventual similarities or differences. 

Line 283: silenced

Lines 289-end: this statement is not supported by sufficient evidences: the picture is of low quality.

Lines 477: review the use of articles

Figure S1 Establishment of BcWRKY33A-... Figure S2: Schematic plot of THE IGSs biosynthetic pathway 

Figure S5: please provide an extended legend: what are TRV?, TRVPDS and TRV MYB51?

Is the plot representing the expression of WRKY33A? What are the error bars? Replicates? Asteriks? Is N/A not analysed or not detected? Is the WRKY expression statistically different n the three experiment (TRV, TRV-PDS, and TRV-MYB)? Use a,b,c lettering to show differences. Since this is not a pairwise comparison, please provide adequate statistics.

Line 326-344 Relevance is not clear.

Line 333: lots of...generic

Line370: sentence meaning unclear. Please revise grammar.

Line371-372: Please revise grammar.

Round 2

Reviewer 2 Report

The authors revised the work as requested, and apported the required changes and clarifications. 

This manuscript is a resubmission of an earlier submission. The following is a list of the peer review reports and author responses from that submission.